# Lineage Reprogramming: Genetic, Chemical, and Physical Cues for Cell Fate Conversion with a Focus on Neuronal Direct Reprogramming and Pluripotency Reprogramming

**DOI:** 10.3390/cells13080707

**Published:** 2024-04-19

**Authors:** Taichi Umeyama, Taito Matsuda, Kinichi Nakashima

**Affiliations:** Department of Stem Cell Biology and Medicine, Graduate School of Medical Sciences, Kyushu University, Fukuoka 819-0395, Japan

**Keywords:** direct reprogramming, pluripotency reprogramming, neuron, epigenetics

## Abstract

Although lineage reprogramming from one cell type to another is becoming a breakthrough technology for cell-based therapy, several limitations remain to be overcome, including the low conversion efficiency and subtype specificity. To address these, many studies have been conducted using genetics, chemistry, physics, and cell biology to control transcriptional networks, signaling cascades, and epigenetic modifications during reprogramming. Here, we summarize recent advances in cellular reprogramming and discuss future directions.

## 1. Introduction

Cellular reprogramming was achieved in the 20th century via nuclear transplantation [1] and cell fusion [2]. These groundbreaking studies demonstrated the existence of intrinsic factors necessary for the conversion of somatic cells and inducing pluripotency, and led to the subsequent development of techniques to reprogram fibroblasts by transducing a minimal set of transcription factors (TFs) to generate induced pluripotent stem cells (iPSCs) (hereafter referred to as pluripotency reprogramming) [3]. More recently, in the last decade, the intrinsic and extrinsic factors required for the direct reprogramming of somatic cells into another differentiated cell type without passing through the pluripotent state have been explored. Recent technological advances such as massive parallel sequencing and single-cell analysis are helping us to better understand the dynamic changes of the transcriptome and epigenome in pluripotency and direct reprogramming. These improvements in lineage reprogramming will be the tailwind for developing more efficient and cell-type-specific reprogramming methodologies.

This review focuses on methodologies for acquiring pluripotency and direct neuronal reprogramming based on (1) genetic cues such as the forced expression of TFs and miRNAs, (2) chemical cues, i.e., small molecules that regulate signal transduction and epigenetic states, and (3) physical cues, including mechanical stimuli and laterally confined cell growth in the culture dish. We would like to emphasize that while these three cues are powerful methods on their own, their synergistic effects can help to develop more efficient and selective cell-type-specific lineage reprogramming technology. General introductions for epigenetics, pluripotent stem cells, neural stem cells, and mammalian gene delivery techniques are listed in Appendix A [4,5,6,7,8]. It may be helpful for readers who are not familiar with these areas.

## 2. Genetic, Chemical, and Physical Cues Affect the Induction and Maintenance of Pluripotency

### 2.1. The Modulation of Signaling Pathways and Epigenetic Statuses Influences the Induction and Maintenance of Pluripotency

In 2006, a major technological breakthrough in science and medicine was made with the report that iPSCs with a gene expression profile and developmental potential similar to those of embryonic stem cells (ESCs) can be generated from mouse somatic cells, fibroblasts, by the forced expression of four TFs, *Oct4*, *Sox2*, *Klf4*, and *c-Myc*, known as the OSKM factors [3]. Subsequently, accumulating evidence has revealed the signaling pathways and gene regulatory networks that govern the induction and maintenance of pluripotency in both ES and iPS cells (Figure 1). In mouse ESCs, the Leukemia inhibitory factor (LIF)-signaling pathway is essential for the self-renewal and maintenance of pluripotency by activating *Klf4* via Janus kinase-signal transducers and activators of transcription 3 (JAK-STAT3) or Tbx3 and its downstream target Nanog via Phosphatidyl inositol 3-kinase-protein kinase B (PI3K-AKT) [9] (Figure 1). However, the LIF shows such beneficial effects only on mESCs derived from species with limited genetic backgrounds, such as 129-inbred mice, because the activation of a pathway other than the LIF-mediated JAK-STAT3 and PI3K–AKT pathways, namely, the Mitogen-activated protein kinase (MAPK) pathway, promotes the differentiation of mESCs [10]. On the other hand, the LIF–STAT3 pathway is dispensable for the maintenance of pluripotency in “primed state” human ES cells [11,12]. Bone morphogenetic protein–small mothers against decapentaplegic (BMP-Smad) (Smad1/5/9) signaling upregulates the inhibitor of differentiation or the inhibitor of DNA-binding (*Id*) genes that, in general, are negative regulators of cell differentiation [13] (Figure 1). These LIF- and BMP-signaling pathways function collaboratively to suppress differentiation and sustain the pluripotency of mouse ESCs [14]. Wingless/Int (Wnt)-β-catenin signaling also plays an important role in the self-renewal and maintaining of the pluripotency of mouse and human ESCs [15]. In mouse ESCs, Wnt-β-catenin signaling reduces active histone marks (H3K4me3 and H3ac) on the promoter region of T-cell factor 3 (Tcf3), which represses the expression of pluripotency genes such as *Nanog*, *Oct4*, and *Tbx3* (Figure 1). Moreover, Wnt-β-catenin signaling induces the upregulation of miR-211, which targets Tcf3. These transcriptional and post-transcriptional repressions of Tcf3 by Wnt-β-catenin signaling regulate the differentiation of mESCs [16]. These facts are well illustrated by the use of MAPK inhibitors and Glycogen synthase kinase-3β (GSK3β) inhibitors, which positively regulate the Wnt–Tcf pathway, in addition to LIF (2i/LIF) (Figure 1), enabling the establishment of naive (also referred to as “ground state”, defined as having the ability to form chimeric mice) ESCs from diverse mouse strains [17]. Several groups have recently also reported the successful establishment of naive human ES and iPS cells by modification based on a 2i/LIF medium for mouse ESCs [18,19]. These facts indicate that a common signaling pathway is vital for the maintenance of pluripotency in mouse and human ES and iPS cells [20].

Another signaling pathway, the Hippo pathway, which responds to mechanical forces from cytoskeletal components, has also been reported to control pluripotency in ES and iPS cells from humans and mice. The Hippo pathway negatively regulates the Yes-associated protein/transcriptional coactivator with the PDZ binding motif (YAP/TAZ) (Figure 1). The YAP and TAZ form a complex with transcriptional enhanced associated domain (TEAD) in the nucleus. This complex reinforces Wnt-signaling activity and induces the expression of cell cycle-related genes and pluripotency TFs such as *OCT4*, promoting self-renewal and sustaining pluripotency for mouse and human ESCs (Figure 1). YAP is activated during iPS-cell reprogramming, and the loss of Large tumor suppressor 1 (LATS1) and LATS2 improves the formation of iPSCs, suggesting the activation of the pluripotent gene network by the YAP in iPSCs [21]. In contrast, the activation of the Hippo pathway by the Mammalian Ste20-like serine/threonine kinase 1/2 (MST1/2) and the LATS1/2 kinase cascades sequesters YAP and TAZ effector proteins in the cytoplasm (Figure 1), resulting in the cytoplasmic accumulation of YAP/TAZ that antagonizes Wnt-signaling activity [22].

Thus, iPSCs generated through pluripotent reprogramming modulate signaling pathways similar to those of ESCs to maintain pluripotency, although the epigenome and transcriptome of ES cells are not identical to those of iPS cells [23].

### 2.2. Genetic Cues Phase-Dependently Affect the Induction of Pluripotency

The process of pluripotency reprogramming is multiphasic, involving dynamic alterations of the epigenome and transcriptome during the acquisition of pluripotency. Gene regulation through histone H3 lysine-4 trimethylation (H3K4me3) and H3K27me3 is observed at CpG-rich promoters, where DNA methylation levels are stably low in somatic cells during reprogramming [24]. On the other hand, gene regulation through DNA demethylation is observed at CpG-poor promoters, where methylation levels are high in somatic cells. Some of these epigenome changes are rapid, while others are slow, which could explain the multiphasic aspect of reprogramming. In accordance with epigenetic alterations, the transcriptome also changes dynamically during the process of pluripotency reprogramming: the first and second waves of transcription changes occur from day 0 to day 3 and after day 9, respectively. Intriguingly, some of these dynamically regulated genes were included in a cocktail of pluripotency-inducing factors, such as *OCT4*, *SOX2*, *NANOG,* and *LIN28* [25], and also *miRNA302/367* [26]. Moreover, the efficiency of OSKM-mediated pluripotency reprogramming is enhanced by the additional expression of factors whose expression is upregulated in the second wave after OSKM induction, for example, the ESC-specific Ras isoform (Eras), Akt coactivator (*Tcl1*), *Nanog*, and *miRNA-302* [27]. Mbd3 (methyl CpG-binding protein 3) is dynamically upregulated during embryonic development and exerts an antagonistic effect on pluripotency reprogramming in the first wave phase (0–5 days after OSKM induction in somatic cells). Using siRNA-mediated Mbd3 knockdown in fibroblasts or B cells, the efficiency of OSKM-induced iPSCs’ generation was boosted to nearly 100% [28]. Given that the manipulation of gene expression with rapid expression changes observed in the first and second wave of the OSKM-mediated reprogramming process contributes to increased reprogramming efficiency, time course analyses of gene expression during pluripotency reprogramming are invaluable for finding genetic cues that boost pluripotency reprogramming.

A recent study also showed that the timing of when to express factors that promote the reprogramming mediated by OSKM is also important. The transient expression of CCAAT/Enhancer-binding protein-α (C/EBP-α) prior to OSKM expression led to a 103-fold increase in iPSC reprogramming efficiency. This was due to the increased Tet2 expression and decreased methylation levels of the *Oct4* promoter. However, this ‘path-breaking’ function of C/EBP-α was not observed when C/EBP-α was expressed after OSKM induction, showing the phase-dependent effect of genetic cues for iPSC generation [29]. Taken together, these findings indicate that the efficiency of pluripotency reprogramming depends not only on controlling the expression of factors that support OSKM function or engineering the OSKM factors [30,31] but also on the timing of their expression.

### 2.3. The Phase-Dependent Effect of Chemical Cues Affects Pluripotency

Small molecules and proteins are effective in generating temporal cues and regulating the intensity and duration of stimuli. Thus, they enhance the efficiency of multi-step initiation processes by regulating signaling pathways and epigenetic states at the appropriate time points before and after the initiation of pluripotency reprogramming. Furthermore, these extrinsic cues can be used in parallel with genetic cues. For example, Histone deacetylase (HDAC) inhibitor valproic acid (VPA) has been utilized to enhance the efficiency of OSKM-induced iPSC generation [32]. Furthermore, because of the high level of Hdac2 in mouse fibroblasts, the Hdac2 protein degradation-promoting activity of VPA is required to generate mouse iPSCs by *miRNA302/367* [26,33]. Oct4-activating compounds (OAC1 and OAC2) enhanced the efficiency of OSKM-induced iPSC generation [34]. Furthermore, one or more OSKM-reprogramming factors can be replaced by chemicals (Figure 1). For example, the Transforming growth factor-β (TGF-β) receptor 1 inhibitor RepSox induces *Nanog* and *L-Myc* expression and enables the generation of iPSCs with only three factors by substituting for either *Sox2* or *c-Myc* [35]. EPZ004777, an H3K79 histone methyltransferase Disruptor of the telomeric silencing 1-like (DOT1L) specific inhibitor [36], decreases H3K79me2 levels during the mesenchymal-to-epithelial transition, functioning as an enhancer and accelerator in the conversion from fibroblasts to iPSCs and also enabling the generation of iPSCs with only the expression of two exogenous genes, *KLF4* and *c-Myc* [37]. Other Dot1L inhibitors (SGC0946 or EPZ5676) increased the binding of SOX2 and OCT4 (or Oct6, 7, 8, 9) to *OCT4* and *NANOG* enhancer regions (Figure 1), providing an epigenetically permissive state for pluripotency reprogramming [38]. *Oct4* overexpression alone was sufficient to generate iPSCs when fibroblasts were treated with several molecules such as protein arginine methyltransferase inhibitor AMI-5 and TGF-β inhibitor A-83-01 [39], BMPs [40], Bmi [41], and the combination of VPA, GSK3-β inhibitor (CHIR99021), RepSox, and the Lysine-specific demethylase 1 (LSD1) inhibitor (Tranylcypromine) [42]. These successful examples of the use of chemical cues for pluripotency reprogramming demonstrate the power of small molecules as surrogates for genetic cues. 

### 2.4. Maximum Efficacy of Pluripotency Induction by Chemical Cues Is Obtained by Optimizing the Timing of Treatment

Non-genetic methods have an advantage in clinical applications because chemical reprogramming methods that are completely independent of exogenous genetic cues can avoid the problems associated with genomic mutations that increase carcinogenicity. Chemically induced pluripotent stem cells (CiPSCs) were generated from fibroblasts by using VPA, CHIR99021, RepSox, Tranylcypromine, adenylyl cyclase activator (Forskolin), retinoic acid receptor agonist (TTNPB), and *S*-adenosylhomocysteine hydrolase inhibitor (DZNep), which induce *Oct4* expression by decreasing DNA and H3K9 methylation at the promoter. However, the conversion efficiency was only 0.2%, and it took 40 days to produce CiPSCs [43]. Recently, a more efficient strategy for generating human CiPSCs utilized chemicals that induce the marker gene expression in each of the four steps of pluripotency reprogramming: stage I (from fibroblast to an epithelial-like state), stage II (from an epithelial-like state to an intermediate plastic state), stage III (from an intermediate plastic state to an extraembryonic endoderm (XEN)-like state), and stage IV (from a XEN-like state to an iPSC) [44]. Guan et al. identified the following chemical compounds as useful for inducing pluripotency reprogramming: VPA(V), CHIR99021(C), RepSox(R), TTNPB (N), Y27632(Y), Smoothened receptor agonist (SAG(S)), receptor tyrosine kinase inhibitor (ABT869(A)), Tranylcypromine(T), DNMTase inhibitor (5-azacytidine(5-aza)), c-Jun N-terminal kinase inhibitor (JNKIN8(J)), EPZ004777(E), MEK inhibitor (PD0325901(P)), SETD8 inhibitor (UNC0379(U)), Wnt/β-catenin-inhibitor (IWP-2(I)), B-Raf inhibitor (SB590885(S)), and DZNep(Z), and these chemicals function in combinations: CRNYSA in stage I, CRNYSATJ5-aza in stage II, CRYPTVEZU in stage III, and CPBIY in stage IV. The additional G9a inhibitor (UNC0224), JAK inhibitor (Ruxolitinib), and CBP/p300 inhibitor (SGC-CBP30) boosted the efficiency of stage II, and the efficacy of CiPSC generation from the adult human fibroblasts was 0.21–2.56% [38], suggesting the utility of time-course analysis to identify chemical cues that induce pluripotency reprogramming. From the optimization of the chemical-reprogramming method, another chemical cocktail consisting of BMP4, B27, Menin–MLL interaction inhibitor (VTP50469), AKT kinase inhibitor (HY-10249A), EPZ5676, casein kinase inhibitor (CX4945), adenosine kinase inhibitor (5-iodotubercidin), and SETD2 inhibitor (EZM0414) was identified, and these small molecules shortened the induction time from approximately 50 days to a minimum of 16 days, and achieved a higher efficiency than OSKM expression using Sendai-virus [45]. Another group also optimized the chemical cocktail in each reprogramming stage and generated CiPSCs in only 12 days. Intriguingly, during the reprogramming, the distribution of H3K9me3 was significantly altered and some H3K9me3-regulated endogenous retroviruses (ERVs) became highly expressed in CiPSCs, and the knockdown of these ERVs decreased the reprogramming efficiency, suggesting that several ERVs are included in the pluripotency network [46]. Taken together, these findings indicate that the efficacy of the chemical induction of pluripotent stem cells is dependent on the timing of treatment, and optimizing the combination of chemicals that are suitable in each phase is indispensable for efficiently generating CiPSCs in a short time.

### 2.5. Physical Cues Generated by the Substrate Surface, Electromagnetic Fields, and Confined Space Affect the Induction of Pluripotency

In physiological conditions, diverse signal transduction pathways are activated by mechanical stimuli such as a high cell density, adherent junctions, liquid flow, and even gravity, and, hence, the nuclear translocation of transcriptional regulators such as YAP and TAZ [47] occurs. Some of these environmental stimuli have been used to improve the efficiency of iPSC generation. For example, fibroblasts adhering to periodic line-patterned microgroove polydimethylsiloxane (PDMS) exhibited an elongated shape that represented cytoskeletal reorganization, causing the upregulation of WD repeat domain 5 (WDR5) H3 methyltransferase and the downregulation of HDAC2, leading to increases of H3K4 di- and tri-methylation and H3 acetylation, and, at least in part, substituting for the treatment with VPA in OSKM-induced iPSC generation [48]. Furthermore, compared with the 2D culture condition, the maintenance of pluripotency and the efficiency of iPSC production from fibroblasts were higher in the 3D microenvironment, which mimics the naïve extracellular matrix by modifying PEG-based hydrogel with adhesion peptides. In the 3D culture condition, YAP1 activity in nuclei was diminished at the onset of reprogramming, and this modulation of YAP1 activity was not observed in the 2D culture condition, indicating that YAP/TAZ signaling may have played a role in the early phase of reprogramming [49].

It has been reported that extremely low-frequency electromagnetic fields (EL-EMFs) enhanced OSKM-induced iPSC generation more effectively than the addition of VPA or vitamin C and that *Oct4* expression alone could produce iPSCs under EL-EMFs [50]. During this process, several characteristic features of pluripotency reprogramming were observed, such as the inhibition of GSK3β and Extracellular signal-regulated kinase 1/2 (ERK1/2), a decreased H3K27me3 level, the increased expression of endogenous *Oct4*, *Nanog*, and *Sox2*, and the DNA demethylation of *Oct4* and *Nanog* promoters. In Oct4/EL-EMF-induced iPSCs, myeloid/mixed-lineage leukemia 2 (Mll2), which is a member of the trithorax group and responsible for H3K4 trimethylation, was upregulated, and the H3K4me3 level was increased not only globally but also in the promoter regions of *Oct4*, *Nanog*, and *Esrrb*. When the magnetic field from the earth was canceled by a Helmholtz coil, OSKM-induced iPSC generation was delayed and Mll2 expression and H3K4me3 levels were decreased [50].

Fibroblasts grown in laterally confined conditions showed an increased promoter occupancy of acetylated H3K9 at the reprogramming genes and more-stem-like transcriptomes as indicated by an increased expression of *Oct4*, *Sox2*, and Wnt pathway-associated genes. These reprogrammed spheroids could be maintained as stem cells in the presence of LIF and also could be differentiated into dopaminergic neuron-like cells expressing βIII tubulin and tyrosine hydroxylase [51], and upon the tuning of the stiffness of the 3D matrix by changing the collagen concentration, the spheroids successfully re-differentiated into the fibroblast-like state. Intriguingly, these fibroblasts showed rejuvenation characteristics including reduced DNA damage and enhanced cytoskeletal gene expression [52], recapitulating iPSC reprogramming-mediated rejuvenation, which is in stark contrast with the retention of age-related signatures in the induced neuronal cells generated by direct reprogramming [53], which is covered in the next section.

## 3. Direct Reprogramming by Genetic, Chemical, and Physical Cues

### 3.1. Direct Reprogramming by Lineage-Specific Factors

Cell lineage conversion technology enabled a switch from one cell type to another without passing through a pluri/multipotent stem cell state (hereafter defined as direct reprogramming). Obtaining the desired cells via direct reprogramming has several advantages compared to iPSC reprogramming, e.g., a short induction period, efficient conversion, reduced risk of tumor formation, and in situ cell lineage conversion at the lesion site without the need for ex vivo culturing. One of the pioneering discoveries of direct reprogramming by the overexpression of a TF is the conversion of mouse embryonic fibroblasts to myoblasts using *MYOD* [54,55], which predated the establishment of iPSCs. Despite its great importance, this achievement had not been paid much attention for a long time but was brought back into the limelight with the advent of iPSCs. To date, several studies have reported the direct reprogramming of fibroblasts to differentiated cells such as cardiomyocytes [56,57,58], hepatocytes [59,60,61], endothelial cells [62], macrophages [63], brown-fat cells [64], Sertoli-like cells [65], and several stem/progenitor cells including hematopoietic progenitor cells [66,67,68], hepatic stem cells [69], cardiovascular progenitor cells [70], skeletal muscle progenitor cells [71], intestinal progenitor cells [72], oligodendrocyte progenitor cells [73,74], and neural progenitor cells (NPCs) [75,76]. These technologies for conversion to diverse cell types have potential clinical applications, and the direct reprogramming of non-neuronal cells to neurons, especially in areas of significant neuronal loss, is expected to open the door to the treatment of diseases of and injuries to the central nervous system. The genetic cues frequently used in neuronal direct reprogramming and induced neuronal subtypes are shown in Figure 2, and the efficacies and expression systems are summarized in Appendix A.

### 3.2. Neuronal Differentiation from Neural Stem Cells in a Physiological Condition

To understand the intrinsic and extrinsic factors that induce neuronal reprogramming, it may be necessary to describe the complex mechanisms that regulate the gene regulatory networks that govern neural stem cells’ (NSCs’) maintenance and differentiation into neurons under physiological conditions. Briefly, LIF signaling through the JAK–STAT pathway induces astrocytic lineage commitment in NSCs [77] (Figure 2A), and the synergistic effect of simultaneous LIF and BMP treatments induces astrocytic conversion by fusing JAK–STAT signaling and BMP–Smad signaling by forming the STAT3–p300–Smad complex, which induces glial fibrillary acidic protein (GFAP), a marker of astrocyte differentiation of NSCs [78] (Figure 2A). The canonical BMP–Smad pathway induces the expression of negative regulators of cell differentiation such as the *Id* and hairy and enhancer of split (*Hes*) genes [79]. In this pathway, BMP receptor kinase phosphorylates R-Smads (Smad1/5/9). The phosphorylated R-Smads interact with Co-Smad (Smad4) and then translocate into the nucleus and induce *Hes*, whose products repress the expression of neuronal genes, including *Ascl1* and *Neurog2* (Figure 2A). The activation of the Notch-Delta signal and the subsequent cleavage events lead to the nuclear translocation of the intracellular domain of Notch (NICD) and induce *Hes* and *Hey* genes, repressing the neuronal differentiation of NSCs [80]. This Notch–Delta pathway and the BMP–Smad pathway synergistically enhance the expression of *Hes* and its close relative *Hesr-1* [81] (Figure 2A). In the canonical TGF-β–Smad pathway, TGF-β receptor kinase phosphorylates the other R-Smads (Smad2/3), which is distinct from the BMP–Smad pathway. The phosphorylated Smad2/3 also associates with Smad4, then translocates into the nucleus, modulating the expression of genes involved in neurogenesis and the exit from the cell cycle [82] (Figure 2A). Wnt signaling induces proliferation and neuronal differentiation in the early and late developmental stages, respectively, showing developmental-stage-dependent effects on the fate decision of NSCs [83]. Wnt–β-catenin signaling is regulated by the phosphorylation-mediated degradation of β-catenin by a protein kinase complex that includes GSK3β, casein kinase, and Axin, and subsequently by the ubiquitin–proteasome pathway [84]. This kinase complex is tethered to the plasma membrane by the binding of the Wnt ligand to the Frizzled receptor, and hence, β-catenin is stabilized and accumulates in the nucleus (Figure 2A). β-catenin suppresses Tcf3, which functions as a suppressor of Wnt signaling and proneural basic helix–loop–helix (bHLH) gene *Neurog1*. During neuronal differentiation, Wnt–β-catenin increases the expression level of the Tcf1 activator, which enhances the Wnt/β-catenin signal itself, leading to positive feedback for neuronal differentiation [85]. In the Sonic hedgehog (Shh)–Glioblastoma associated oncogene (Gli) pathway, membrane receptor Patched (Ptch) inhibits the activity of the Smoothened (Smo) receptor in the absence of Shh. Gli2 and Gli3 (Gli2/3) TFs are phosphorylated by a protein kinase complex including GSK3β, casein kinase, and protein kinase A (PKA), and then Gli2/3 are cleaved and converted to the repressor form (Figure 2A). Shh binding to Ptch derepresses the activity of Smo and prevents proteolytic cleavage, and then full-length active Gli2/3 induces Gli1 and hedgehog target genes that encode factors involved in cell proliferation, survival, and self-renewal [86]. The fibroblast growth factor (FGF) pathway mainly consists of Ras–MAPK-, PI3K–AKT-, and PLCγ–PKC-signaling pathways [87], and plays an important role in NSC survival and proliferation (Figure 2A). Both the Ras–MAPK and PI3K–AKT pathways are involved in self-renewal of NSCs [88]. However, PLCγ signaling has a distinct role and is crucial to the maintenance of stemness [89]. Several isozymes of PKCs such as classical (α, β1, β2, and γ), novel (δ, ε, θ, and η), and atypical (ζ and λ) are expressed in NSCs, and FGF signaling requires PKCδ [90], and these isozymes are relevant to diverse functions and often determine growth factor specificity [91]. Extracellular environmental factors such as stiffness and properties of the extracellular matrix, pressure, and shear stress are sensed by mechanosensors. Integrins are one of the primary mechanosensors and trigger RhoA activity and Rho kinase (ROCK). Activation of Rho–ROCK and the downstream pathway increases the filamentous actin (F-actin)-to-globular actin (G-actin) ratio and polymerized actin fibers activate the TFs YAP and TAZ, which drive numerous genes, including genes affecting cell proliferation [92] (Figure 2A).

### 3.3. Direct Neuronal Reprogramming from Fibroblasts by Genetic Cues

The conversion into neurons was historically studied first using cultured cells in vitro, and fibroblasts, in particular, have been widely used, as they are readily available from the body. The overexpression of single TF *ASCL1* generated only immature neurons with inadequate neurite growth, and co-culture with glial cells was necessary for functional maturation [93]. Thus, *Ascl1* alone is not sufficient to induce mature functional neurons, and additional neuronal maturation-promoting TFs are needed to generate functional neurons. For example, *Ascl1* together with *Brn2* (*Pou3f2*) and *Myt1l* (ABM condition) generated GABAergic neurons from fibroblasts [94] (Figure 2B) and even from hepatocytes [95]. In the ABM condition, Ascl1 acts as a pioneer TF and binds to the trivalent chromatin state (H3K4me1, H3k27ac, and H3K9me3), then recruits Brn2 [96] and regulates other key TFs, including Zfp238, Sox8, and Dlx3 [97]. In addition to ABM factors, *NeuroD1* overexpression in human fibroblasts enhanced neuronal conversion 2–3 fold [98], and ABM together with *Isl1*, *Lhx3*, *Neurog2,* and homeobox gene *Hb9*, which encodes a motor neuron determinant, converted fibroblasts into motor neurons [99] (Figure 2B). Furthermore, additional *FoxA2* and *Lmx1a* (ABMFL) converted human fibroblasts to functional induced dopaminergic neurons (iDAs) [100] (Figure 2B), and *Lmx1b* and *Otx2* increased the efficiency of this process [101]. 

Although *Brn2* and *Myt1l* were required to induce functional neurons in the ABM condition, they are not essential factors in other conditions. For example, *Ascl1* together with *Foxg1*, *Lhx6*, and *Sox2* could convert fibroblasts to GABAergic interneurons [102] (Figure 2B). *Pitx3*, which was shown to be required for the generation of mature iDA from ESCs [103], induced only immature iDAs from fibroblasts when used with *Ascl1*. However, *Pitx3* and *Ascl1* together with an additional four factors (*Lmx1a*, *Nurr1*, *Foxa2*, and *EN1*) were capable of producing mature iDAs from fibroblasts (Figure 2B). Interestingly, *Lmx1b* could not substitute for *Lmx1a* in the production of mature iDA [104], although *Lmx1b* is known to be involved in physiological dopaminergic differentiation [105]. Other studies also succeeded in generating iDAs via the forced expression of *Ascl1*, *Lmx1a*, and *Nurr1* in vitro [106], and transplanting these iDAs ameliorated the symptoms of Parkinson’s disease model rats generated with neurotoxin 6-hydroxydopamine injection [107]. Taken together, these results suggest that the ability of combinations of TFs to induce neuronal subtypes from fibroblasts is context-dependent.

Three functional classes of peripheral sensory neurons (nociceptors, mechanoreceptors, and proprioceptors) that selectively express the Trk receptor subfamily, TrkA, TrkB, and TrkC, were generated by the overexpression of *Brn3a* with either *Neurog1* or *Neurog2* in fibroblasts (Figure 2B). These induced sensory neurons evoked action potentials and responded to several ligands known to activate pain- and itch-sensing neurons, showing functionally similar characteristics to native peripheral sensory neurons [108]. Another combination of TF genes (*Ascl1*, *Brn2*, *Myt1l*, *Neurog1*, *Isl2*, and *Klf7*) also generated nociceptor neurons (Figure 2B), but, intriguingly, the addition of *Brn3a* expression markedly inhibited reprogramming [109]. These findings further support the notion that the neuronal conversion from fibroblasts induced by the expression of TF genes is context-dependent. In addition to TFs, miRNAs play important roles in neuronal differentiation and are also used in neuronal direct reprogramming. Although treatment with only *miR-9/9** and *miR-124* could convert human fibroblasts into neurons (Figure 2B), the efficiency of this process was boosted by *ASCL1* and *NEUROD2* synergistically [110]. In the neuronal differentiation mentioned above, *Ascl1* is mostly indispensable. The ABM condition has been widely used in many neuronal direct reprogramming studies, but *miRNA-124* together with *BRN2* and *MYT1L* also succeeded in converting fibroblasts to glutamatergic and GABAergic neurons [111] (Figure 2B). The simultaneous expression of *miR-9/9**, *miR-124*, *BCL11B*, *DLX1*, *DLX2*, and *MYT1L* converted fibroblasts to striatal medial spiny neurons [112], showing the potential of these miRNAs to increase the number of tools that could be chosen for direct reprogramming. Thus, neuronal direct reprogramming using fibroblasts as starting cells mostly requires the proneural gene *Ascl1*, which encodes a bHLH TF, miRNAs, or both [93,94,96,97,98,99,100,101,102,103,106,107,109,110,111,112]. Until now, an increasing number of successful examples of direct reprogramming have been reported, but only a few studies have performed a time-course analysis to unveil the cellular state transition during the processes of cell fate conversion. The cell lineage trajectories of many direct reprogramming studies induced by the genetic cues mentioned above will help us to develop more efficient and cell-type-specific neuronal reprogramming strategies.

### 3.4. Chemical Cues Boost the Efficiency of Direct Neuronal Reprogramming

Small molecules that modulate signal transduction pathways have a large potential to enhance the efficiency of direct reprogramming and have been used to generate diverse cell lineages. For example, while the treatment of *Oct4*-overexpressing fibroblasts with a combination of chemical compounds (CHIR99021, Forskolin, Tranylcypromine, and TGF-β inhibitor [SB431542]) induced cardiomyocyte conversion [113], the treatment of the cells with another combination (TGF-β inhibitor [A83-01], CHIR99021, LPA, phosphodiesterase 4 inhibitor [Rolipram], and JNK inhibitor [SP600125]) resulted in their conversion into NSCs [114]. In the field of neuronal direct reprogramming, the neuronal conversion efficacy by genetic manipulation could be boosted and oriented to some specific subtypes by distinct combinations of compounds. Forskolin enhanced the efficacy of conversion from human fibroblasts to electrophysiologically functional cholinergic motor neuron-like cells (hiMNs) induced by *Neurog2*, and a BMP signal inhibitor (Dorsomorphin) supported cell survival in this process [115]. These hiMNs were capable of forming neuromuscular junctions with skeletal muscle, but hiMNs generated from amyotrophic lateral sclerosis (ALS) patients could not. However, such degenerative features of hiMNs from ALS patients could be rescued by GSK-3β inhibitor kenpaullone [116]. Neuronal conversion from pericytes via a stem cell-like state was induced by the overexpression of *Ascl1* and *Sox2* [117], and this process was inhibited by Nodal or BMP4 but induced by SB431542 and Dorsomorphin, and also induced by γ–secretase inhibitor N-[N-(3,5-difluorophenacetyl)-L-alanyl]-S-phenylglycine t-butyl ester (DAPT) by disrupting the NOTCH-signaling pathway [118]. These successful examples of chemically assisted direct reprogramming show the utility of small molecules as enhancers of direct reprogramming and strongly indicate that further studies are needed to explore small molecules that selectively modulate the signal transduction pathway.

### 3.5. Chemical Direct Reprogramming without Genetic Cues

Since non-genetic direct reprogramming arouses less concern about carcinogenesis; chemical direct reprogramming methodologies are favorable for clinical applications and have already been used to generate several cell lineages. For example, human cardiomyocyte-like beating cells were generated in 6–8 days via treatment with chemicals previously shown to be useful for generating for CiPSCs, which was much shorter than the time required to generate CiPSCs (20 to 40 days) [119]. Such cardiomyocyte-like beating cells were also generated by another chemical combination (CHIR99021, A83-01, G9a inhibitor (BIX01294), KDM5B inhibitor (AS8351), ERK1 and Ras GTPase inhibitor [SC1], ROCK inhibitor [Y27632], OAC2, PDGFR-β inhibitor [SUF16F], and PDGFR-α/β inhibitor [JNJ10198409]) [120]. In the chemical cocktail used for the generation of CiPSCs, DZNep and forskolin induce the expression of *Oct4*, and when these two compounds were substituted by OAC1, and used together with VPA, CHIR99021, SB431542, and tranylcypromine, mouse embryonic fibroblasts were converted to functional astrocytes [121]. 

In another chemical direct neuronal reprogramming, human fibroblasts were converted to βIII tubulin-positive neurons largely composed of vGLUT1-positive glutamatergic neurons via treatment with VPA, CHIR99021, RepSox, forskolin, SP600125, Y-27632, and PKC inhibitor (GO6983) [122]. Other chemical cocktails consisting of forskolin, CHIR99021, BET bromodomain inhibitor (I-BET151), and Wnt/β-catenin agonist (Isoxazole-9 or ISX9) efficiently convert mouse fibroblasts to neurons, and I-BET151 represses fibroblastic genes and ISX9 induces neuronal genes, respectively, partly unveiling the mechanism of neuronal conversion during this process [123]. Photoreceptor-like cells were generated via the treatment of fibroblasts with VCRFI (VPA, CHIR99021, RepSox, Forskolin, and Wnt/β-catenin inhibitor [IWR1]) and the subsequent addition of Shh, taurine, and retinoic acid [124]. During this process, VCRFI stabilizes AXIN2 and it translocates to mitochondria. This translocation results in the generation of reactive oxygen species and the subsequent activation of Nuclear factor-κB (NF-κB), increasing the ability of *Ascl1* to induce photoreceptor-like cell conversion from fibroblasts. The transplantation of the chemically induced photoreceptor-like cells was capable of partially restoring the function of vision in mice. The experimental conditions used for chemical reprogramming mentioned in this section are summarized in Appendix A.

### 3.6. Direct Reprogramming by Physical Cues

Cell-lineage conversion is also induced by mechanical signals such as the structure and stiffness of the surface contacting the cells, extracellular force, and electromagnetic fields. These physical cues can be used in parallel with genetic and chemical cues, and they are therefore expected to become useful tools for direct reprogramming. Mechanoregulation by YAP/TAZ plays an important role in neuronal development [125], innate immune response [126], stem cell fate decision [21], and tumor growth by epigenetic remodeling [127,128,129]. The differentiation of human mesenchymal stem cells depends on the substrate stiffness, and soft matrices that mimic the brain are neurogenic, stiffer matrices that mimic muscle are myogenic, and rigid matrices that mimic bone are osteogenic [130]. The substrate stiffness activates and enhances the nuclear localization of YAP/TAZ [131]. As expected from these studies, YAP and/or TAZ were exported from the nucleus by soft matrices in the direct reprogramming process from fibroblasts to neurons [132] and also cardiomyocytes [133]. 

In addition to substrate stiffness, substrate structure also affects the efficacy of neuronal conversion. In combination with treatment with the genetic cues *Ascl1*, *Pitx3*, *Nurr1*, and *Lmx1a* in fibroblasts, a nanogrooved surface with a 400 nm width increased genome-wide H3K4me3 levels compared with a flat substrate and increased the efficiency of converting fibroblasts to functional dopaminergic neurons [134]; and a microgrooved surface with a 10 μm width converted fibroblasts to αMHC-positive cardiomyocytes more efficiently than the condition using an overexpression of mechanosensitive TF Mkl1 and treatment with VPA on a flat surface [135]. The large-scale screening of micro/nanopatterns on the surface identified several unconventional hierarchical patterns as efficient physical cues for direct neuronal reprogramming. Using an optimal surface structure, the chemical conversion efficiency from human fibroblasts to βIII tubulin-positive cells was boosted from 17% to 74% [136].

Gold nanoparticles (AuNGs) covered by RGD heptapeptide (CYGRGD), which binds integrin, were exposed to a specific EMF frequency and became transiently magnetized. Use of the magnetized AuNGs together with the transfection of *Ascl1*, *Pitx3*, *Lmx1a*, and *Nurr1* (APLN) improved the conversion efficiency of fibroblasts to iDAs approximately 6-fold compared to the EMF-free condition [137]. In addition, only the presence of EMF, CHIR99021, Forskolin, and VPA successfully generated βIII tubulin- and Map2-positive neuronal cells. In APLN/EL-EMF-induced iDA, chromatin-modifying bromodomain-containing protein-2 (Brd2), which is a key regulator of histone acetylation, was upregulated, and H4K12 and H3K27 acetylation levels were increased genome-wide. Also, EMF increased H3K27 acetylation at the transcription start site (TSS) region of several neuronal genes including *Lmx1a*, *NeuroD1*, *DAT*, *NeuN*, *Nf1*, *Med12*, and *Hpn*, and increased H4K12 acetylation at the TSS region of dopaminergic neuronal marker gene tyrosine hydroxylase. EMF/AuNGs also enhanced the conversion rate of human fibroblasts to electrophysiologically functional iDAs, and significantly ameliorated the symptoms of Parkinson’s disease model mice when AuNG was injected together with APNL into the striatum in the presence of EMF [137]. The elongation of the nucleus by the passing of mouse fibroblasts through a 7-μm microchannel induced the deformation of the nucleus and enhanced the direct neuronal conversion efficiency of the fibroblasts in the ABM condition. This squeezing condition significantly increased the features of mature neurons in the cells, i.e., Tau-, βIII tubulin-, and Map2-positive cells, compared with the transgene-only condition, and also increased these cells’ calcium fluctuation. In these “squeezing iNs”, lamin A/C showed transient wrinkling and a decrease at the nuclear periphery, and H3K9me3 and 5mC levels were globally decreased. Although YAP translocation into the nuclei was not affected by squeezing, the knockdown of mechanosensitive ion channel Piezo-1 slightly decreased reprogramming efficiency. The same squeezing-mediated reprogramming also boosted the efficiency of the conversion of macrophages to neurons in the ABM condition, and the conversion of fibroblasts to iPSCs by OSKM, suggesting the generality of squeezing-mediated physical cues for boosting reprogramming efficiencies in diverse applications [138]. Despite lots of knowledge about mechanosensory signal transduction systems, the mechanism of reprogramming by physical cues is not well understood, especially in signal transduction systems. Therefore, unveiling the mechanism of reprogramming, together with the further hunting of physical cues that affect cell fate conversion processes as one desires, will help to develop this research area into a mature field of cell biology and take it closer to clinical applications.

### 3.7. Direct Neuronal Reprogramming from Glia

Direct neuronal reprogramming from glia is expected to provide a novel therapeutic strategy for neural injuries and neurodegenerative diseases. After neural injury, astrocytes, NG2 glia, ependymal cells, and microglia become activated, proliferate, and migrate to the injured sites [139] and participate in the formation of a glial scar. These cells are beneficial in the acute phase by restricting damage spreading, but cause detrimental effects on recovery by limiting the axon extension of neurons in the chronic phase, and are therefore adverse prognostic factors for neuronal regeneration. For this reason, glial cells in and surrounding the injured site are regarded as promising sources of direct neuronal reprogramming for treating neural injuries.

The direct neuronal reprogramming from astrocytes, one type of glial cells, was reported before the advent of iPSCs. Firstly, neuronal conversion was accomplished by the overexpression of only a single TF *Pax6* [140], and also by the forced expression of *Ascl1* or *Neurog2* [141]. The transcriptomes of these *Ascl1*-induced and *Neurog2*-induced neurons obtained from astrocytes were distinct from each other, and only a small subset of genes, including pan-neuronal genes, were shared. Among them, *Neurod4*, which is regulated by the repressor element-1 silencing TF (REST) complex in competition with Neurog2, was capable of reprogramming astrocytes into electrophysiologically functional neurons [142]. Dlx2 is a member of the homeobox TF Dlx family which is necessary for differentiation into GABAergic neurons [143]. As expected from this property of these family members, GABAergic neurons were generated from astrocytes by the overexpression of *Dlx2* [144]. In physiological conditions, *Lmx1b* and *Nurr1* are involved in dopaminergic neuronal differentiation and survival, respectively [105,145], and the overexpression of these two genes together with *ASCL1* generated induced dopaminergic iDAs from astrocytes [146].

NeuroD1 could convert astrocytes to glutamatergic neurons, and also convert polydendrocytes (NG2 cells) to glutamatergic and GABAergic neurons [147]. However, in this study, human microglia failed to convert into neurons, and the authors mentioned the low infection efficiency of microglia by retroviruses as a reason for the failure. Indeed, using a lentivirus infection system, mouse microglia were successfully reprogrammed into neurons by *NeuroD1* overexpression [148], and this process tightly depends on the *NeuroD1* expression level [149]. In the conversion process, NeuroD1 binds to the bivalent domain containing both active H3K4me3 and repressive H3K27me3 modifications, and neuron-specific genes were upregulated before the downregulation of microglia-specific genes, suggesting the importance of time-course analysis to elucidate the reprogramming process. As mentioned above, glial cells have several advantages as the starting cells for neuronal direct reprogramming, and therefore pursuing in vivo neuronal reprogramming is worthwhile to improve functional recovery after nervous tissue injury.

### 3.8. Direct In Vivo Reprogramming for Treatment of Neural Injury

In addition to in vitro reprogramming, in vivo neuronal reprogramming from glial cells has recently been increasingly studied, and improved neural function in a mouse model of disease has been reported. For instance, astrocytes in adult mice striatum transduced with a genetic cue, *SOX2,* with lentivirus, re-entered into the cell cycle and became DCX-positive neuroblasts [150]. Furthermore, a chemical cue, VPA, promoted the differentiation of these induced neuroblasts into electrophysiologically active mature neurons. Similarly, the ectopic expression of *SOX2* transduced with lentivirus in astrocytes after spinal cord injury (SCI) induced DCX-positive neuroblasts and the VPA-induced maturation of these cells, forming synaptic contacts with endogenous motor neurons. These results indicated that a combination of genetic and chemical cues enables in vivo functional neuronal reprogramming. Importantly, only *SOX2* could induce DCX-positive neurons among the tested 12 genes (*SOX2*, *PAX6*, *NKX6.1*, *NEUROG2*, *ASCL1*, *OLIG2*, *SOX11*, *Tlx*, *OCT4*, *c-Myc*, *KLF4*, and *PTF1a*) and the strong expression of *SOX2* in the initial reprogramming process and its subsequent downregulation was required for the reprogramming [151]. In addition, the SOX2-mediated reprogramming to neuroblasts in the spinal cord was promoted by the downregulation of p53 and p21, which mediate cell cycle arrest [152]. Single TF-mediated in vivo neuronal reprogramming from astrocytes was also accomplished using TFs other than Sox2. As mentioned above, although Dlx2 was reported to induce GABAergic neurons from astrocytes in vitro [145], the lentivirus-mediated transduction of Dlx2 reprogrammed astrocytes in adult mouse striatum to multiple lineage NPCs and generated not only neurons but also astrocytes and oligodendrocytes [153]. Thus, the results of cell fate conversion by the same genetic cues are different in vitro and in vivo. The reason for this is that the chemical and physical stimuli surrounding the cells are totally different. Furthermore, the temporal changes of such stimuli in vivo may also affect the gene expression induced by genetic cues. Adeno-associated virus (AAV)-mediated *Neurod1* expression in the ischemic injured cortex converted astrocytes to NeuN-positive neurons, which increased the cortical tissue volume at the injured site, sent out their axon projections to distant target areas even through the corpus callosum, and recovered cognitive fear memory [154]. NG2 glial cells were also used as starting cells for in vivo neuronal direct reprogramming. For example, NG2 glia in adult mouse striatum transduced with *Ascl1*, *Lmx1a,* and *Nurr1* with AAV were converted to both excitatory and inhibitory neurons, and these induced neurons made synaptic contact with endogenous neurons, showing the integration with the local host neuronal circuit [155]. Moreover, the ectopic expression of *SOX2* in NG2 glia after SCI generated excitatory and inhibitory mature neurons with a cell lineage passing through proliferating neuroblasts. These induced neurons formed synaptic connections with endogenous neurons and improved locomotor recovery after spinal cord injury, increased the neurons surrounding the lesion, and decreased the volume and surface area of the glial scar [156]. These studies using mouse models show the great potential of in vivo reprogramming using glia as starting cells for future clinical applications of direct reprogramming to the treatment of neural injury.

## 4. Conclusions

Since the first discovery of myogenic reprogramming, genetic cues such as the overexpression of TFs and miRNAs have been explored to generate cells showing appropriate functions for target cells such as specific neuronal subtypes, myocytes, hepatocytes, and even pluripotent stem cells. As observed in the change between the functions of Wnt/β-catenin signaling in the early and late stages of neuronal differentiation in the physiological condition, functions of genetic cues are assumed to change in the course of reprogramming, so the most effective timing of a genetic cue’s triggering may be different in each case. Technological advancements and cost reduction in transcriptome and epigenome analysis have enabled us to analyze the trajectories of cell lineage conversion and understand the discrepancy between induced cells and the desired target cells at each time point of interest [157], so that more efficient and subtype-specific reprogramming methods can be developed by changing the genetic cues at an appropriate time point to direct the transcriptome and epigenome to the desired states. Furthermore, in cells expressing the pluripotency-reprogramming factor OSKM and neuronal-reprogramming factor NeuroD1, the large-scale modification of factors that may promote cell fate conversion and scRNA-seq can reveal combinations of previously unknown genetic cues that effectively induce pluripotency and neuronal reprogramming [158].

Although these technological advances have boosted the development of procedures for reprogramming by genetic cues, genetic manipulation unavoidably entails the possibility of increasing the oncogenic potential. Therefore, mimicking the effect of genetic cues with chemical cues will be the second stage of developing a reprogramming technology. Traditional basic biology approaches, such as exploring the effect of small molecules on modulating signal transduction systems and transcriptional networks, are increasingly important for developing novel therapeutic strategies by cellular reprogramming with chemical cues.

Compared with genetic and chemical cues, physical cues are non-invasive and spatiotemporally controllable. The manipulation of the 3D microenvironment by physical cues may have a great advantage for developing reprogramming methodologies for generating complex cellular systems consisting of multiple cell lineages such as organoids. However, although mechanosensory transduction has been studied extensively and many extracellular environments, including substrate stiffness and structure, shear stress, and even gravity, are already known to influence gene expression, only a limited number of physical cues have so far been shown to affect cellular reprogramming. Therefore, further studies will be required to explore the physical cues that modulate the cellular state. In addition, physical cues have begun to be applied to enhance the efficiency of the direct reprogramming and iPSC generation induced by genetic and chemical cues. We believe that making full use of genetic, chemical, and physical cues together with the further exploration of other cues such as bacterial infection [159,160] will help us to develop a more sophisticated methodology to generate cells with desired functions like those of naturally observed cells and/or even cells with desirable functions that are not possessed by any cell types existing in nature.

## Figures and Tables

**Figure 1 cells-13-00707-f001:**
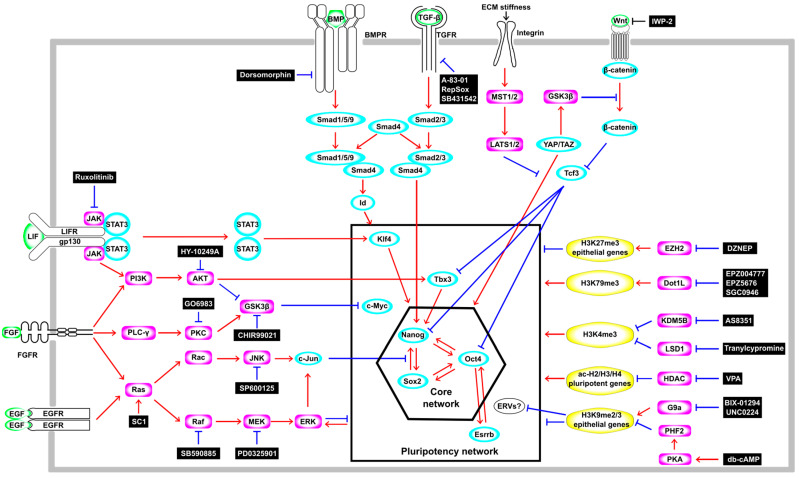
Schematic illustration of the signal transduction cascade, the transcription network, and epigenetic regulation in the regulation of pluripotency. Ligands are shown in green, transcription factors are shown in cyan, enzymes are shown in magenta, and histone modifications are shown in yellow.Red arrows indicate activation and blue blunt arrows indicate inhibition.

**Figure 2 cells-13-00707-f002:**
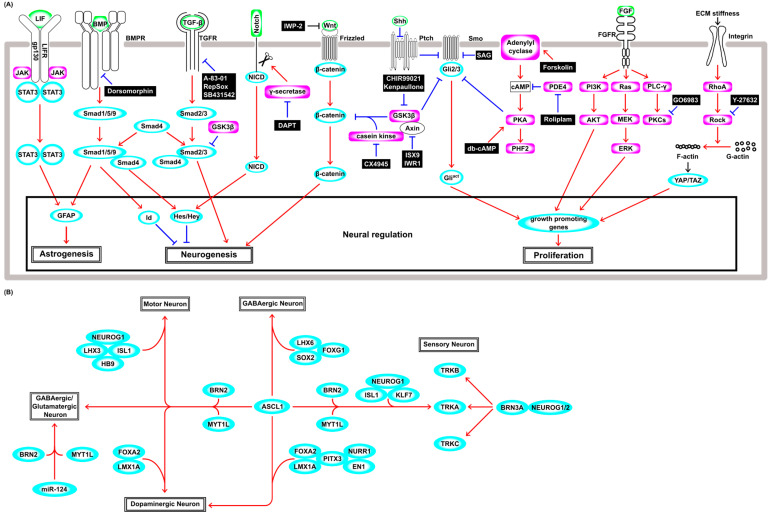
A schematic illustration of neuronal regulation. (**A**) The signal transduction cascade regulating the astrogenesis, neurogenesis, and cell proliferation. Ligands are shown in green, transcription factors are shown in cyan, and enzymes are shown in magenta. Red arrows indicate activation and blue blunt arrows indicate inhibition. (**B**) The transcription network regulating direct neuronal reprogramming from fibroblasts. Transcription factors and miRNAs are shown in cyan. Red arrows indicate activation.

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
