# Peer review of "Lineage Reprogramming: Genetic, Chemical, and Physical Cues for Cell Fate Conversion with a Focus on Neuronal Direct Reprogramming and Pluripotency Reprogramming"

_cells, 2024, doi:10.3390/cells13080707_

Round 1

Reviewer 1 Report

Comments and Suggestions for Authors

This work is helpful as data about induced pluripotent stem cells and cell re programming is of much current interest.  This manuscript summarises signalling pathways and molecules that are used in these processes as well as different factors that can influence cell differentiation. There is a lot of information and I feel a list of abbreviations would be helpful. I also think readers who are not familiar with this area might find it helpful if references / reviews that provide basic introductory background could be listed.  I also note that some of the experimental findings in the laboratory are not similar to those that occur in embryology or tumour development in vivo. For example, matrix in tumours is believed to be formed by the tumour cells (e.g., osteoid), rather than matrix influencing tumour cell differentiation.  More discussion about the differences in experimental conditions and the in vivo situation would be of interest. There are occasional typos e.g., Yes- associated protein is listed as Yeas- associated protein on page 2 line 79, so that the authors would need to check quite thoroughly as there are many technical data in this manuscript.

Reviewer 2 Report

Comments and Suggestions for Authors

The manuscript by Umeyama, Matsuda and Nakashima presents a review on cellular reprogramming focusing on reprogramming to pluripotency and neuronal cells. Numerous reviews have been written on these topics, however up-to-date reviews appear meaningful on this relevant biomedical research area.

In general the review is well-written and gives a balanced overview on the to be covered topics. Figure 1 and 2 are very detailed and give a good overview and insights into regulation of pluripotency and neuronal regulation. I could not detect any flaws in the review - of course the focus on some cited work over others is always debatable.

I just have some minor points, that should be considered in the manuscript.

The difference between mouse and human pluripotent cells concerning the LIF pathway should be pointed out. Daheron et al. (2004) (10.1634/stemcells.22-5-770) and several further studies are pointing that LIF/STAT3 signaling is not working in human pluripotent stem cells similar as in mouse. However, the 2i/LIF combination when establishing ‘naïve’ human PSC, points at functionality under these circumstances.

I would cover / cite the Boyer et al. (2005) (10.1016/j.cell.2005.08.020) which is to my awareness the first study to point at the conservation of the OSN pluripotency core network between mouse and human, what predicts that OSKM reprogramming is conserved between mouse and human.

One recent aspect to be included in paragraph 2.2. (TF-induced pluripotency) should be to not only use combinations of ‘natural’ transcription factors like OSKM, but also to design ‘artificial’ transcription factors, which boost reprogramming efficiency and quality of iPSC e.g. as in Tan et al. (2021) (10.1093/molbev/msab075), MacCarthy et al. (2024) (10.1016/j.stem.2023.11.010).

On the neuronal reprogramming part of the review, paragraphs 3.1.-3.3./3.7.-3.8. cover aspects which have been reviewed in multiple manuscripts.  The parts 3.4.-3.6. cover currently underrepresented approaches.

In general, I am in favor of publication in Cells with the minor points to be addressed.
